# AN UNFORGEABLE PUBLICLY VERIFIABLE WATERMARK FOR LARGE LANGUAGE MODELS

**Aiwei Liu[1], Leyi Pan[1], Xuming Hu[3], Shu'ang Li[1], Lijie Wen[1] ***,
**Irwin King[2], Philip S. Yu[4]**

[1]Tsinghua University    [2]The Chinese University of Hong Kong
[3]The Hong Kong University of Science and Technology (Guangzhou)
[4]University of Illinois at Chicago
`liuaw20@mails.tsinghua.edu.cn, wenlj@tsinghua.edu.cn, king@cse.cuhk.edu.hk`

## ABSTRACT

Recently, text watermarking algorithms for large language models (LLMs) have been proposed to mitigate the potential harms of text generated by LLMs, including fake news and copyright issues. However, current watermark detection algorithms require the secret key used in the watermark generation process, making them susceptible to security breaches and counterfeiting during public detection. To address this limitation, we propose an unforgeable publicly verifiable watermark algorithm named UPV that uses two different neural networks for watermark generation and detection, instead of using the same key at both stages. Meanwhile, the token embedding parameters are shared between the generation and detection networks, which makes the detection network achieve a high accuracy very efficiently. Experiments demonstrate that our algorithm attains high detection accuracy and computational efficiency through neural networks. Subsequent analysis confirms the high complexity involved in forging the watermark from the detection network. Our code is available at https://github.com/THU-BPM/unforgeable_watermark.

## 1 INTRODUCTION

With the development of current large language models (LLMs), many LLMs, like GPT-4 (OpenAI, 2023) and Claude, could rapidly generate human-like texts. This has led to numerous risks, such as the generation of a vast amount of false information on the Internet (Pan et al., 2023), and the infringement of copyrights of creative works (Chen et al., 2023). Therefore, texts generated by LLMs need to be detected and tagged.

At present, some watermarking algorithms for LLM have proved successful in making machine-generated texts detectable by adding implicit features during the text generation process that are difficult for humans to discover but easily detected by the specially designed method (Christ et al., 2023; Kirchenbauer et al., 2023). The current watermark algorithms for large models utilize a shared key during the generation and detection of watermarks. They work well when the detection access is restricted to the watermark owner only. However, in many situations, when third-party watermark detection is required, the exposure of the shared key would enable others to forge the watermark. Therefore, preventing the watermark forge in the public detection setting, is of great importance.

In this work, we propose UPV, the first unforgeable publicly verifiable watermarking algorithm for LLMs. Our approach adopts the commonly used watermark schema, which embeds small watermark signals into LLM's logits during generation Kirchenbauer et al. (2023); Zhao et al. (2023). The main difference is that our algorithms could detect the watermark without the generation key. To conceal the details of the watermark in the detection process, we propose using two separate neural networks for watermark generation and detection, rather than relying on the same secret key for both stages. The unforgeability of our algorithm stems from a computational asymmetry: constructing a watermark generation network from a watermark detection network is notably more complex than the reverse process (Section 4.5). To enhance the accuracy of the detection network while alleviating

---

*Corresponding author

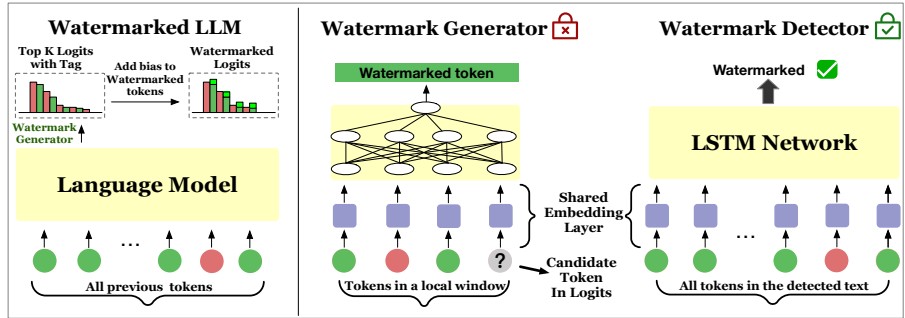

Figure 1: The left segment outlines the token logits generation process of the watermarked language model. Initially, the origin model generates token logits; these are then refined by the watermark generator to increase the probability of watermarked tokens (denoted in green). Operating within a predefined token window, this generator network (center) determines the final token's watermark label. The watermark detector (right) evaluates the entire text to ascertain watermark presence.

the complexity of the training process, we have shared the token embedding parameters between the detector and the generator. This provides some prior information to the detector while maintaining unforgeable. Specifically, our watermark generator takes a window of $w$ tokens as input and predicts the watermark signals of the last token. The text detector directly takes all tokens from the text as input and outputs whether the complete text contains the watermark.

In our experiments, the watermark detection algorithm exhibits a detection performance of nearly 99% F1 score, almost identical to the key-based watermark algorithm, which is our theoretical upper bound. Notably, the computational burden of our watermark generation and detection network is minimal, especially when compared to large language models, due to its significantly lower parameter count. Furthermore, we have demonstrated the difficulty of cracking the watermarking rules to forge watermark through detailed experiments. We find that both attacking strategies, employing a detector to construct training data for the generator training and using word frequency analysis, yield exceedingly low rates of successful decryption in scenarios with not exceptionally small window size.

The main contributions of this work can be summarized as follows:

- Introduction of a novel unforgeable publicly verifiable algorithm named UPV that employs separate neural networks for watermark generation and detection.
- Utilization of shared token embedding between the watermark generation and detection networks, thereby improving the efficiency of training the detector.
- Through empirical analysis, we have demonstrated that our watermarking algorithm is highly resistant to attacks trying to forge the watermark.

## 2  RELATED WORK

LLMs have been successfully applied to many tasks (Liu et al., 2023a; Hu et al., 2023b). However, the potential for misuse of LLMs still remains. To mitigate these misuses, strategies such as red teaming (Liu et al., 2022) and safety alignment (Ji et al., 2023) can be employed. Nonetheless, not all forms of misuse can be addressed through these methods, making it imperative to detect and track text generated by LLMs. Currently, there are two primary methods for identifying LLM-produced text: text watermarking and classifier-based detection.

Classifier-based approaches typically fine-tune LLMs like GPT-2 (Radford et al., 2019), BART(Lewis et al., 2019), and RoBERTa (Liu et al., 1907) for binary text classification (Zhan et al., 2023; Mireshghallah et al., 2023). Some methods incorporate language model features (Su et al., 2023) or adversarial training (Hu et al., 2023a). However, the fundamental detectability of machine-generated text remains an open question. Some argue that with sufficient data collection, high-performing detectors are achievable (Chakraborty et al., 2023). Sadasivan et al. (2023) contend that as LLMs become more advanced, the performance gap between detectors and random classifiers may diminish.

Compared to the classifier-based methods, text watermarking (Liu et al., 2023c) is more explainable. There are typically two categories of text watermarking methods. The first is to add a watermark to the existing text. For example, Abdelnabi & Fritz (2021) designed a data-hiding network to embed watermark information in the text, and utilized a data-revealing network to recover the embedded information. Yoo et al. (2023) injected the watermark by substituting some words in the text. However, adding a watermark to the existing text struggles to keep the semantics of the text unchanged which limits its use in real-world scenarios. Another line of methods is injecting the watermark during the text decoding process. Christ et al. (2023) used a pre-selected value sequence to sample the tokens and subsequently detected the watermark by observing the correlation between the preset pseudorandom numbers and the generated tokens. The work of Kuditipudi et al. (2023) introduced Levenshtein distance to measure the distance between generated text and random number sequences, improving the robustness of watermarks. Kirchenbauer et al. (2023) divided the vocabulary into red and green lists and preferred to generate tokens from the green list. Zhao et al. (2023) enhanced the robustness of this approach by using a global fixed red-green vocabulary. Liu et al. (2023b) achieve a balance between robustness and security through semantic invariant hashing. Lee et al. (2023) designed a watermarking method for low-entropy code generation scenarios. However, the above methods require the key from the watermark generation phase during the detection, which exposes the watermark's key when made accessible for public detection. In this work, we propose the first unforgeable publicly verifiable watermark to alleviate these issues.

## 3 PROBLEM DEFINITION

**Language Model:** A language model, denoted by $\mathcal{M}$, is fundamentally a function for predicting the next token in a sequence. $\mathcal{M}$ takes an input sequence $\boldsymbol{x} = [x_0, \ldots, x_{n-1}]$ and produces the probability distribution $P_n$ over a vocabulary set $\mathcal{V}$. Formally, we have $P_n = \mathcal{M}(\boldsymbol{x}_{0:n-1})$. The next token is selected from the distribution, either via sampling or other methods.

**Watermarking Algorithm** comprises watermark generation and watermark detection. `Watermark Generation:` This can be seen as a subtle modification to a language model's probability distribution. If $\hat{\mathcal{M}}$ denotes a watermarked language model, the token prediction probability is expressed as: $\hat{P}_n := \hat{\mathcal{M}}(\boldsymbol{x}_{0:n-1})$. `Watermark Detection:` This algorithm takes a text sequence $\boldsymbol{x} = [x_0, \ldots, x_n]$ and determines the presence of a watermark. There's a one-to-one correspondence between the watermark detection model, denoted as $\mathcal{D}$, and the watermarked language model $\hat{\mathcal{M}}$.

**Unforgeable Publicly Verifiable Watermark:** If a watermarking algorithm is publicly verifiable, it implies that the watermark detector is accessible to the public or third parties, allowing everyone to acquire details of the watermark detection algorithm. When a publicly verifiable watermark algorithm also satisfies the characteristic of being unforgeable, it signifies that even if the public gains access to the watermark detector, it remains challenging to get the watermark generation method (the watermark key), thereby hindering the ability to forge the watermark.

**Threat Model:** For potential attackers, there are two primary attack scenarios. The first involves attempts to modify watermarked text, employing techniques such as text rewriting or synonym substitution to eliminate the watermark. This attack assumes that the attacker does not have access to the detector or similar models, as such access would facilitate easy removal through multiple trials. If a user can consistently remove the watermark (over 90%) using a rewriting algorithm, then the watermark is considered broken. The second scenario arises when the attacker has access to the detector and attempts to decipher the watermark's generation rule for forgery purposes. In this situation, the attacker can access the detector and make unlimited queries. Defenders can only ensure security by designing unforgeable watermark algorithms to prevent forgery. If an attacker gains enough understanding of the watermark generation rule (over a threshold) to easily forge watermarked text, the watermark is considered "broken". In the context of watermarking, two texts are different meaning they are labeled differently (watermarked or not) or convey different meanings.

## 4 PROPOSED METHOD

This section provides a detailed description of our unforgeable publicly verifiable watermark algorithm. The watermarked LLM decoding step is first explained (Section 4.1). We then detail the

watermark generation network (Section 4.2) and principles of watermark detection (Section 4.3 and Section 4.4 ). Finally, we analyze the unforeability of the algorithm (Section 4.5).

## 4.1 WATERMARKED LARGE LANGUAGE MODEL

Algorithm 1 describes how to generate watermarked text with a target language model $\mathcal{M}$. Given an input sequence $\boldsymbol{x} = [x_0, \ldots, x_{n-1}]$, $\mathcal{M}$ produces logit scores $P_n$ for the next token. Afterwards, the candidate tokens will be processed by the watermark generation network $\mathcal{W}$ to produce the watermarked token set G. Logit scores for tokens in $G$ are incremented by $\delta$, while others remain unchanged, forming the logits of the modified model $\hat{\mathcal{M}}$. As is shown in Equation 1.

Not all tokens in the logits are labeled during each generation step. In the case of top-$K$ sampling, only the highest-ranking $K$ tokens are tagged. During beam search with a beam size of $B$, tokens are labeled only if their scores surpass the $B$th highest score $S_B$ by a margin of $\delta$, formally defined as $\{x_i \in \mathcal{V} \mid P_n^{(i)} > S_B - \delta\}$. A more detailed analysis is provided in the appendix H.

---

**Algorithm 1** Watermark Generation Step

1: **Input:** Watermark generation network: $\mathcal{W}$. Fixed integer: $K$. Watermark strength: $\delta$. Language model: $\mathcal{M}$. Previously generated text sequence: $\boldsymbol{x} = [x_0, \ldots, x_{n-1}]$. Local window size: $w$.
2: Compute logits $P_n$ of token $x_n$ given $\boldsymbol{x}_{0:n-1}$: $P_n = \mathcal{M}(\boldsymbol{x}_{0:n-1})$.
3: Extract the top-$K$ tokens from logits, denoting them as $\boldsymbol{x}_n^K = \mathsf{topK}(P_n)$.
4: Initialize sequence matrix $\mathbf{S}$ where each row corresponds to the sequence $[x_{n-w+1}, \ldots, x_n^{(i)}]$ for every $x_n^{(i)} \in \boldsymbol{x}_n^K$.
5: Calculate the watermark result $\boldsymbol{r} = \mathcal{W}(\mathbf{S})$, where each entry $\boldsymbol{r}_i$ is 0 ($x_n^{(i)}$ is not watermarked) or 1 ($x_n^{(i)}$ is watermarked).
6: Generate the watermarked token set $G$ containing items with value 1 in $\mathbf{r}$: $G = \{x_n^{(i)} | \boldsymbol{r}_i = 1\}$.
7: Construct an augmented language model $\hat{\mathcal{M}}$ such that for an input sequence $\boldsymbol{x} = [x_0, \ldots, x_{n-1}]$, the adjusted logits are:

$$\hat{\mathcal{M}}(\boldsymbol{x}_{0:n-1})^{(i)} = \mathcal{M}(\boldsymbol{x}_{0:n-1})^{(i)} + \delta \mathbf{1}(i \in G), \tag{1}$$

where $\mathbf{1}(\cdot)$ is an indicator function: it returns 1 if $i \in G$ and 0 otherwise.
8: **Output:** Modified language model $\hat{\mathcal{M}}$.

---

## 4.2 WATERMARK GENERATION NETWORK

The watermark generation network's architecture is depicted in the middle part of Figure 1. The shared embedding network, denoted as $E$, first generates the embedding for each input token. Subsequently, embeddings from a local window $w$ are concatenated and processed by the fully connected classification network to ascertain if the last token in the watermarked token set. We use FFN to denote the fully connected network and the process could be represented as follows:

$$\mathcal{W}(\boldsymbol{x}_{n-w+1:n}) = \mathrm{FFN}(\mathrm{E}(x_{n-w+1}) \oplus \mathrm{E}(x_{n-w+2}) \oplus \ldots \oplus \mathrm{E}(x_n)). \tag{2}$$

The embedding network accepts the binary representation of token IDs as input. The required number of encoding bits is contingent upon the vocabulary size. For instance, GPT2 Radford et al. (2019) has a vocabulary size $|\mathcal{V}|$ of 50,257, requiring 16 bits for its binary representation.

For effective watermark detection, it's imperative that the proportion of watermarked tokens generated by the generation network remains invariant. Specifically, for any local window $[x_{n-w+1}, \ldots, x_n]$, the probability that $x_n$ belongs to the watermarked token set should consistently be $\gamma$:

$$\forall [x_{n-w+1}, \ldots, x_{n-1}], P(\mathcal{W}([x_{n-w+1}, \ldots, x_{n-1}, x_n]) = 1) = \gamma, \tag{3}$$

where the $\gamma$ has the same meaning as the green list ratio in the previous key-based watermark algorithms Kirchenbauer et al. (2023); Zhao et al. (2023).

Due to the inherent complexity of neural networks, maintaining a static ratio via predefined parameters is non-trivial. We address this challenge by curating a training dataset maintaining the precise

proportion $\gamma$. This approach guarantees an expected watermarked token ratio $\gamma$ with standard deviation $\sigma$. Section 4.3 shows the minimal influence of $\sigma$ on the detection. More training details of the watermark generation network could be seen in the appendi C.

## 4.3 WATERMARK DETECTION

In this section, we introduce how to detect a given watermark using the z-value test. Given a vocabulary partitioned into watermarked and non-watermarked tokens based on a fixed ratio, $\gamma$, the expected number of tokens from the watermarked set in a standard text of length $T$ is $\gamma T$, with a variance of $\gamma(1 - \gamma)T$. Using the z-value test method as proposed by Kirchenbauer et al. (2023), we reject the null hypothesis and detect the watermark in text if the z-score below surpasses a threshold:

$$z = (|s|_G - \gamma T)/\sqrt{T\gamma(1 - \gamma)}. \tag{4}$$

However, as discussed previously, our watermark generation network does not ensure a constant ratio $\gamma$. Instead, a ratio $\hat{\gamma}$ is achieved with an expected value $\gamma$ and a standard deviation $\sigma$. This necessitates a modification of the earlier formula. While the expected count of the green tokens remains $\gamma T$, the variance must be adjusted. The updated variance can be expressed as:

$$Var(\gamma T) = E[Var(\gamma T|\gamma)] + Var(E[\gamma T|\gamma]) = \gamma(1 - \gamma)T + \sigma^2 T, \tag{5}$$

and the new z-score could be calculated as follows:

$$z = (|s|_G - \gamma T)/\sqrt{\gamma(1 - \gamma)T + \sigma^2 T}. \tag{6}$$

Since our standard deviation $\sigma$ is very small in practice, the increase in variance, $\sigma^2 T$, is also quite minimal. In the process of subsequent experiments, we will initially estimate the variance of the generation network and then include the variance during the z-score test calculation.

## 4.4 WATERMARK DETECTION NETWORK

The z-value test effectively detects watermarks in text but requires knowing the label (watermarked or not) of each token during detection. This allows the watermark to be more easily removed or forged. To address this, we propose a watermark detection neural network that accepts only the text sequence as input and outputs whether it contains a watermark.

The detailed structure of our watermark detection network is illustrated in the right part of Figure 1. The input to the entire network is the ID sequence of all tokens in the target sentence, where an output of 1 indicates the presence of a watermark in the entire sentence, and 0 signifies its absence.

Specifically, all tokens first pass through a shared embedding network. Notably, the parameters of this token embedding are consistent with those of the watermark generation network and remain frozen during subsequent training. The motivation behind this novel approach is the shared embedding could give prior information to the detection networks and substantially reduce the difficulty of training.

The token embeddings are then combined and fed into an LSTM network Hochreiter & Schmidhuber (1997) for binary classification of whether the text contains a watermark:

$$\mathcal{D}(\boldsymbol{x}) = \text{LSTM}(\text{E}(x_0) \oplus \text{E}(x_1) \oplus .... \oplus \text{E}(x_T)). \tag{7}$$

The watermark detection network functions as a discriminator, judging if the z-value of input text exceeds a threshold. We construct the training data using equation 6 with a predefined threshold $z$.

Notably, the input of the detection network need not be meaningful text; any token ID list suffices. Thus, the randomly generated ID sequence as training data theoretically avoids out-of-domain issues.

To impede attackers from inferring watermarking rules, we treat the text as a cyclic document, labeling initial tokens based on trailing ones. This results in random labels for the first few tokens, minimizing their effect on detection given the small window relative to full text length. Examples of the cyclic document and training details of the detection network are shown in the appendix F and C.

## 4.5 ANALYSIS OF THE UNFORGEABILITY

To further demonstrate the unforgeability of our watermarking algorithm, this section provides a detailed analysis of the difficulty in cracking the watermark generation method.

Table 1: Empirical detection result for watermark detection using top-K sampling and beam search. Each row is averaged over $\sim 500$ generated sequences of length $T = 200 \pm 5$. The table compares our proposed network-based watermark detection method and the key-based watermark detection that directly calculates the z-score. The hyperparameters are uniformly set as $\delta = 2.0, \gamma = 0.5$.

| Methods / Dataset | | | C4 | | | DBPEDIA CLASS | | |
|---|---|---|---|---|---|---|---|---|
| | | | FPR | FNR | F1 | FPR | FNR | F1 |
| GPT2 | Top-K Sample | Key-based | 0.2 | 0.4 | 99.7 | 0.2 | 0.0 | 99.9 |
| | | Network-based (UPV) | 0.2 | 1.1 | 99.3 | 0.6 | 1.4 | 99.0 |
| | Beam Search | Key-based | 0.2 | 0.2 | 99.8 | 0.2 | 0.0 | 99.9 |
| | | Network-based (UPV) | 0.2 | 0.7 | 99.6 | 0.1 | 0.8 | 99.6 |
| OPT 1.3B | Top-K Sample | Key-based | 0.0 | 0.4 | 99.8 | 0.0 | 0.6 | 99.7 |
| | | Network-based | 0.2 | 4.7 | 97.5 | 0.8 | 3.1 | 98.0 |
| | Beam Search | Key-based | 0.0 | 0.0 | 100.0 | 0.0 | 0.0 | 100.0 |
| | | Network-based (UPV) | 0.2 | 0.6 | 99.6 | 0.8 | 0.4 | 99.4 |
| LLaMA 7B | Top-K Sample | Key-based | 0.0 | 1.0 | 99.5 | 0.0 | 3.0 | 98.5 |
| | | Network-based (UPV) | 0.4 | 3.2 | 98.2 | 1.2 | 3.7 | 97.5 |
| | Beam Search | Key-based | 0.0 | 0.0 | 100.0 | 0.0 | 0.6 | 99.7 |
| | | Network-based (UPV) | 0.1 | 0.4 | 99.8 | 0.5 | 0.9 | 99.3 |

In the process of watermark generation, we use watermark generation networks to produce labels for training watermark detection networks. This process is straightforward. For a given text $\boldsymbol{x}$, the gold label of the detection network $\mathcal{D}$ is a simple function (denoted as f) of watermark generation network:

$$\mathcal{D}(\boldsymbol{x}) = f\left(\mathcal{W}(\boldsymbol{x}_{0:w}), \mathcal{W}(\boldsymbol{x}_{1:w+1}), ..., , \mathcal{W}(\boldsymbol{x}_{T-w:T})\right). \tag{8}$$

Conversely, utilizing the watermark detection network to produce training labels for the watermark generation network presents substantial challenges. Given the input text $\boldsymbol{x}$ and the detection network $\mathcal{D}$, it is not straightforward to obtain a specific label $\mathcal{W}(\mathbf{x}_{i:i+w})$. The inference is only possible with respect to the relationships among multiple labels. This necessitates accounting for output interdependencies within the generation network, as shown below with a more complex function $g$:

$$\mathcal{W}(\boldsymbol{x}_{i:i+w}) = g\left(\boldsymbol{x}, \mathcal{D}, \{\mathcal{W}(\boldsymbol{x}_{j:j+w})\}_{j \neq i}\right). \tag{9}$$

The uncertainty and complex dependencies between labels make training watermark generation networks from watermark detection networks extremely difficult. Details about the complex nature of this process will be expanded upon in the experiment section. This characteristic adequately satisfies computational asymmetry, which is vital for unforgeability-preserving fields like asymmetric cryptography. The theoretical training difficulty and computational asymmetry inherent in training the generation network from the detection network robustly safeguard the unforgeability.

In addition to using the watermark detection network to train the generation network, one can infer the watermark rules by analyzing word frequency differences across large amounts of text, i.e. the spoofing attack mentioned by Sadasivan et al. (2023). The principle is that when using a watermarking method with a window size $w$, the frequency of the $w$th token with fixed previous $w - 1$ tokens will differ from that in unwatermarked text. However, this approach is largely ineffective when the window size is not particularly small, as will be detailed in the experiments section.

## 5 EXPERIMENT

### 5.1 EXPERIMENT SETUP

**Language Model and Dataset:** We utlize three language models - GPT-2(Radford et al., 2019), OPT-1.3B(Zhang et al., 2022), and LLaMA-7B(Touvron et al., 2023) - to generate watermarked text. Consistent with previous works, we employ two decoding methods: Top-K sampling and Beam search, to produce the watermarked text. We select the C4(Raffel et al., 2020) and Dbpedia Class (Raffel et al., 2020) datasets and use the first 30 words of each text as the prompt. For each prompt, the language model would generate the next 200 ± 5 tokens.

Table 2: The table presents an ablation study on the shared layer, contrasting the effects of using a shared layer (w. shared layer), not using a shared layer (w.o. shared layer), and with shared layer after fine-tuning (w. ft shared layer).

| Methods / Datasets | | C4 | | | DBPEDIA CLASS | | |
|---|---|---|---|---|---|---|---|
| | | FPR | FNR | F1 | FPR | FNR | F1 |
| GPT2 | w. shared-layers | 0.2 | 1.1 | 99.3 | 0.6 | 1.4 | 99.0 |
| | w/o shared-layers | 0.0 | 99.76 | 0.5 | 0.9 | 68.1 | 47.5 |
| | w ft shared-layers | 0.2 | 16.2 | 90.9 | 0.6 | 13.4 | 92.5 |
| OPT 1.3B | w. shared-layers | 0.2 | 4.7 | 97.5 | 0.8 | 3.1 | 98.0 |
| | w/o shared-layers | 0.0 | 100.0 | 0.0 | 0.0 | 99.4 | 1.3 |
| | w ft shared-layers | 0.0 | 22.3 | 87.3 | 0.2 | 25.9 | 84.8 |
| LLaMA 7B | w. shared-layers | 0.4 | 3.2 | 98.2 | 1.2 | 3.7 | 97.5 |
| | w/o shared-layers | 1.2 | 81.0 | 31.4 | 42.0 | 11.4 | 76.9 |
| | w ft shared-layers | 0.2 | 21.3 | 87.8 | 1.0 | 31.3 | 79.6 |

**Evaluation:** The objective in evaluating watermarking algorithms is to distinguish human-written text from language model-generated text. Specifically, we select 500 texts from the dataset as human text and 500 language model-generated texts for evaluating binary classification metrics. We thoroughly document the false positive rate, false negative rate and F1 score. About baselines, we use key-based method as detecting watermark by calculating the z-score with the generation network, and network-based detection to denote our method.

**Hyper-parameters:** The default hyperparameters are configured as follows: watermark token ratio $\gamma$ of 0.5, window size $w$ of 5, five token embedding layers, and $\delta = 2$ for the generator. The detector comprises two LSTM layers as well as the same token embedding layers. For decoding, Top-K employs $K = 20$ and Beam Search utilizes a beam width of 8. Both networks adopt a learning rate of 0.01, optimized via the Adam optimizer (Kingma & Ba, 2014). Due to the varying sensitivity of language models to the watermark, for ease of comparison, the detection z-score thresholds for GPT-2, OPT 1.3B, and LLaMA 7B are set to 1, 1 and 3, respectively.

## 5.2 MAIN RESULTS

Table 1 shows a comparison of detection efficacy between our network-based watermarking detection algorithm and current key-based watermarking. The key-based baseline watermarking detection algorithm is the method of Kirchenbauer et al. (2023), which directly computes the number of watermarked (green) tokens and then calculates the z-score.

As illustrated in Table 1, similar to the key-based watermarking detection algorithm, our network-based watermarking detection algorithm also scarcely produces false positive results (0.1% and 0.4% on average), meaning that human text is almost never mistakenly identified as watermarked text. Moreover, in most scenarios, the false negative probability is only marginally higher than the key-based watermarking detection algorithm by an average of 1.2%. Considering that the performance of the key-based watermarking detection algorithm represents the strict upper bound of our method, this is indeed an outstanding result. We also find that the performance trends of our algorithm across different settings (language models, datasets) are similar to the key-based watermarking detection algorithm. This demonstrates the general applicability of our network-based watermarking detection method without using watermark generation rules. We will analyze the difficulty of inferring watermark generation details from the detector in subsequent analyses.

## 5.3 ANALYSIS OF SHARED EMBEDDING

To demonstrate the necessity of shared embedding layers between the generation network and detection network, we compared the watermark detection performance under three settings in Table 2: without using shared embedding layers, using shared embedding layers but not fine-tuning them during detection network training, and using shared embedding layers while also fine-tuning them during detection network training. All the experiments employ top-K sampling.

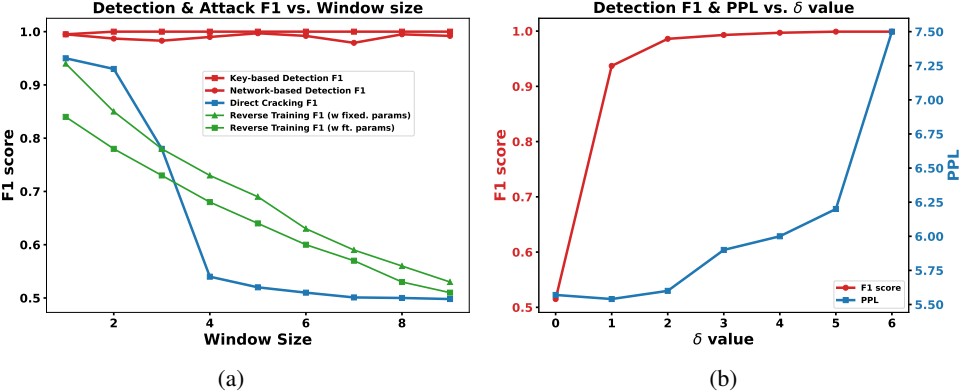

(a)  (b)

Figure 2: The left figure shows the detection success rate of our unforgeable publicly verifiable watermark and the success rate of two attack algorithms on the watermark under different window sizes. The right figure shows how watermark detection F1 score and generated text quality (measured by text perplexity) change as $\delta$ increases.

Without the shared layer, F1 score decreases dramatically, by 72.0% on average. This renders the detection algorithm nearly useless. Furthermore, fine-tuning the shared embedding layers decreases F1 score by 11.1%. These results demonstrate using the generator's token embedding layers without further fine-tuning is optimal for the detection network.

### 5.4 UNFORGEABILITY ANALYSIS

It is crucial that the details of watermark generation remain difficult to infer. Thus, we conducted an analysis of the detection F1 score of potential attack methods aimed at discerning the watermark generation rules. Specifically, we examined two attack methods mentioned in Section 4.5: reverse training of the generation network from the detection network, and direct cracking of the watermark based on token frequency. For the reverse training generation network from the detection network attack method, the retrained generation network also shares parameters in the token embedding layers of the detection network without further fine-tuning. The setting *w_ft. params* indicates that the token embedding parameters have been fine-tuned during the training of the detection network or and the setting *w fixed. params* indicates the parameters are the same as the origin generation network.

From Figure 2(a), our watermark detection algorithm can maintain a relatively stable detection F1 score as the window size increases. However, the effectiveness of the attack method based on reverse training decreases gradually as the window size increases until it drops to the minimum value of 0.5 (random guess). This strongly validates our explanation in Section 4.5 that training the generator with the detector is a computational asymmetrical inverse process. At the same time, the cracking method directly based on word frequency is completely useless when the window size is not particularly small. This experiment demonstrates that our method has good unforgeable properties. More details about the attack algorithms could be seen in the appendix D.

### 5.5 HYPER-PARAMETER AND ERROR ANALYSIS

In investigating the efficacy of our unforgeable publicly verifiable watermark algorithm, we focus on the influence of the $\delta$ values on the detection F1 score and the text quality. Text quality is assessed by perplexity using the LLaMA 13B (Touvron et al., 2023). Figure 2 (b) demonstrates that as $\delta$ increases, so does the F1 score of the detection network; however, this comes at the cost of an increased perplexity (PPL) value in the generated text. After considering these trade-offs, we selected a $\delta$ value of 2 to optimize the detection F1 score without severely compromising text quality.

To analyze the error cases, we present the z-score distributions of human and watermarked texts, as well as the algorithm's detection F1 score at different z-score ranges in Figure 3(a). These results are generated by LLaMA-7B on the C4 dataset. Figure 3(a) reveals that the human and watermarked

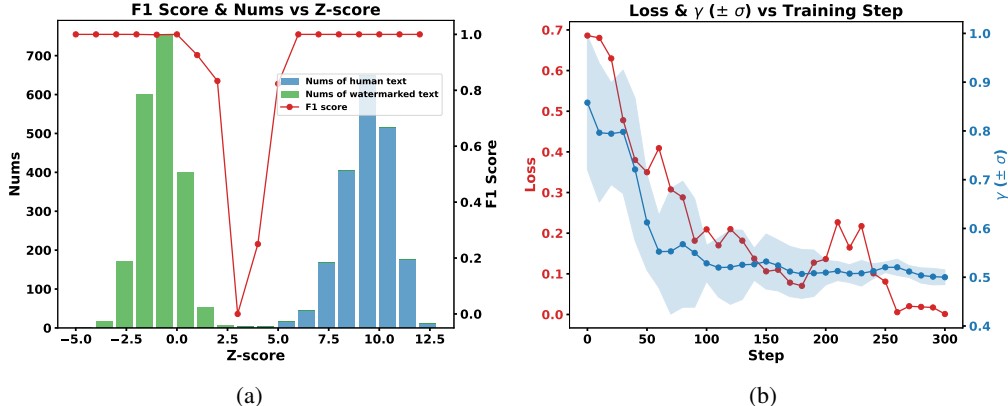

(a) (b)

Figure 3: The left figure is an error analysis, illustrating the detection F1 score for data within various ranges of z-scores. The right figure depicts the changes in loss and the mean proportion ($\pm$ standard deviation) of watermarked tokens generated by the watermark generator network during training.

texts exhibit nearly normal distributions around -1 and 9, respectively. The detection F1 score of the our watermark algorithm drops significantly around the z-score threshold of 3 but approaches 100% in other ranges. This indicates our algorithm is highly reliable for inputs with definite labels.

### 5.6 WATERMARK GENERATION NETWORK ANALYSIS

It is critical that the watermark generation network produce a stable watermarked token ratio, as the modified z-score calculation (equation 6) depends on the variance of this ratio. Therefore, in this section, we calculate the actual mean and variance of the watermark labels.

Specifically, we train the watermark generation network using 5000 data items with strictly a 0.5 ratio of watermarked tokens. As seen in figure 3 (b), the ratio approaches the target value 0.5 as the training loss decreases, and its standard deviation also diminishes. The standard deviation can be controlled within 0.02, corresponding to a variance of less than $4e-4$. According to equation 6, $\sigma^2 T$ could be nearly neglected in the final z-value calculation.

### 5.7 TIME COMPLEXITY ANALYSIS

Our watermark generation process requires an additional network, potentially introducing computational overhead. However, our watermark generation network contains only 43k parameters, negligible compared to the 124M to 7B parameters in GPT2, OPT-1.3B, and LLaMA-7B models. Empirically, decoding a token takes 30ms in GPT2 on a single Tesla V100 GPU, and our watermarking adds only 1ms on average, even less for larger models. Thus, our watermark generation algorithm adds minimal computational burden. In prior experiments, we selected relatively small top-K values. However, given that networks can perform batch computations, even large K values do not affect the computation time. A detailed analysis will be provided in the appendix H.

## 6 CONCLUSION

In this paper, we propose the first unforgeable publicly verifiable watermarking algorithm for large language models. Unlike previous detection methods that require the watermark key for detection, our method uses a separate detection neural network to detect the watermark. In experiments, we demonstrate the difficulty of inferring the watermarking method from the detection network, while our method achieves similar F1 scores to direct z-score computation. In the appendix B, we demonstrate that our watermarking algorithm remains robust against a single instance of text rewriting. However, its robustness in the face of multiple rewrites or more intense attack scenarios remains unexplored. This aspect falls beyond the scope of our current work, and future research could focus on enhancing the robustness of the unforgeable publicly verifiable watermarking algorithm.

## 7 ACKNOWLEDGMENTS

This work is supported by the National Nature Science Foundation of China (No. 62021002), Tsinghua BNRist, and the Beijing Key Laboratory of Industrial Bigdata System and Application. Additionally, it receives support from the Beijing Natural Science Foundation under grant number QY23117. Moreover, the work is partially supported by a grant from the Research Grants Council of the Hong Kong Special Administrative Region, China (CUHK 14222922, RGC GRF No. 2151185). This work is also supported in part by NSF under grant III-2106758.

At the same time, we would like to express our gratitude to the anonymous ICLR reviewers PMgN, CjAu, seaW, and kypk, as well as the Area Chair vQPR, for their valuable feedback, which played a significant role in improving the quality of our paper. We especially appreciate reviewer CjAu, who helped us clarify the definition of our methods and provided many valuable suggestions for revisions, greatly enhancing the quality of our paper.

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

Table 3: Selected output examples from non-watermarked (NW) and watermarked (W) top-K sampling using $\gamma = 0.5$, $\delta = 2.0$ and $k = 20$.

| prompt | real completion | no watermark (NW) | watermarked (W) | (NW) z | (W) z | (Real) PPL | (NW) PPL | (W) PPL |
|---|---|---|---|---|---|---|---|---|
| DPR members Jim Ragsdale, Diane Kane, Angeles Liera and pro tem chair Mike Costello discuss condo conversion projects.\n During the Jan | .10 meeting of the La Jolla Development Permit Review committee (DPR), board members voted unanimously to form a research subcommittee that will look into the consequences of condo conversion in the neighborhoods south of Pearl Street.[...continues] | . 27 meeting, Ragsdale and Kane discussed the need for a condo market and how to get there. Costello spoke about the need to have a condo market in the area but also said there is a need to be able to rent a condo and that the area is growing. [...continues] | . 11-12 meeting, the city announced that the development of condos to be built in the historic downtown has been approved by the city.\n We feel it's important to be able to provide affordable housing for the people of the city in a way that the community feels they [...continues] | 0.71 | 10.5 | 5.42 | 8.26 | 7.15 |
| In their first game since dropping out of the top five, the Irish delivered a redemption performance against Boston College, picking up a 50-point win over | the Eagles while simultaneously moving one step closer to cementing Arike Ogunbowale's legacy, as the senior guard passed current associate coach Beth Cunningham on the list of all-time scorers in the program.\n No. 6 Notre Dame (23-2, 10-2 ACC) wasted [...continues] | the Eagles. The Irish also defeated the Bulldogs in the final and will face the Bulldogs in the final.\n The Irish also defeated the Eagles in the final and will face the Bulldogs in the final. Boston College:\n The Eagles had a very good game against the Irish, [...continues] | South Carolina in the College Football Playoff Tournament to secure a berth in the NCAA Tournament. The Eagles will meet Notre Dame in the Big 12 Tournament on Sept. 14 in Austin, Texas.\n We've got to win in the first round [...continues] | 1.13 | 11.3 | 4.78 | 6.78 | 9.15 |
| Two officers of a company that operates three Manhattan hotels were indicted yesterday in a scheme to help homeless people fraudulently obtain welfare checks and split the money | with the hotels.\n District Attorney Robert M. Morgenthau of Manhattan, who announced the indictments, said they resulted from a study of the three hotels announced last December by the city. He said his office was delayed in moving more quickly on the case because of difficulty[...continues] | they received from the government.\n The indictment, which was released on Tuesday, said that police officers, who arrived at the hotel on a routine shift, met with a homeless man who asked for a check from his landlord. man told the officers about the scheme, which involved[...continues] | between themselves and the homeless.\n The scheme, alleged to be connected to the New York City Department of Health and Welfare, was uncovered in the wake of the 2012 Sandy Hook Elementary School shooting and the 2011 bombing of the Boston Marathon. Authorities say that the scheme [...continues] | -1.9 | 9.94 | 4.83 | 7.02 | 7.05 |
| Buddhadev had written a strong letter of protest to Manmohan Singh objecting to Mulford's behaviour. \n Taking serious exception to | US Ambassador David Mulford writing directly to West Bengal Chief Minister Buddhadev Bhattacharjee for his remarks against the American President, the CPI-M on Friday said the party[...continues] | the comments made by Mulford, the BJP MP also called on the CM to resign immediately and the Centre to make a statement in the coming weeks.\n In his letter to Manmohan Singh, the MP said he was not opposed to[...continues] | Mr Mulford's behaviour in the media and in the Parliament, the Union Minister has directed the Union Secretaries of Parliament and the Secretaries of the Supreme Court to take action against him in the matter[...continues] | 1.85 | 12.36 | 4.25 | 7.35 | 8.15 |

Table 4: This table presents the detection accuracy of our algorithm in text rewriting scenarios. We compared the performance of our network-based watermark detection algorithm with the previous key-based watermarking detection methods under the C4 dataset, both in the absence of attacks and after rewriting using GPT-3.5.

| Methods / Settings | | | NO ATTACK | | | REWRITE W. GPT3.5 | | |
|---|---|---|---|---|---|---|---|---|
| | | | FPR | FNR | F1 | FPR | FNR | F1 |
| GPT2 | w. window_size 1 | Key-based | 0.0 | 0.0 | 100.0 | 2.4 | 4.7 | 96.4 |
| | | Network-based | 0.6 | 0.0 | 99.7 | 6.4 | 1.0 | 96.4 |
| | w. window_size 2 | Key-based | 0.0 | 0.0 | 100.0 | 4.6 | 9.3 | 92.8 |
| | | Network-based | 0.0 | 1.0 | 99.5 | 5.0 | 2.6 | 96.2 |
| OPT 1.3B | w. window_size 1 | Key-based | 0.0 | 0.0 | 100.0 | 6.8 | 8.9 | 92.0 |
| | | Network-based | 0.0 | 0.0 | 100.0 | 17.4 | 3.4 | 90.3 |
| | w. window_size 2 | Key-based | 0.0 | 0.0 | 100.0 | 7.8 | 11.5 | 90.1 |
| | | Network-based | 0.4 | 1.0 | 99.3 | 4.6 | 11.2 | 91.8 |
| LLaMA 7B | w. window_size 1 | Key-based | 0.0 | 0.0 | 100.0 | 8.4 | 12.5 | 89.0 |
| | | Network-based | 0.0 | 0.0 | 100.0 | 18.0 | 5.0 | 89.2 |
| | w. window_size 2 | Key-based | 0.0 | 0.0 | 100.0 | 11.0 | 13.7 | 87.2 |
| | | Network-based | 1.8 | 1.0 | 98.6 | 12.4 | 2.6 | 92.9 |

# A  CASE STUDY

To better illustrate the text generated by the watermarked LLM, we have listed some text examples from both the watermarked LLM and the non-watermarked LLM in Table 3. We compare the z-scores and PPL scores between these texts. Specifically, when calculating PPL scores, we utilize the LLaMA 13B model (Touvron et al., 2023). The results from table 3 demonstrate that the z-scores for texts generated by the watermarked LLM are significantly higher than those from the non-watermarked LLM, while there isn't a significant increase in the PPL scores.

## B  ROBUSTNESS TO REWRITE ATTACK

To further illustrate the robustness of our watermarking method, we compared our approach with a key-based watermark detection algorithm in Table 4, focusing on the detection F1 scores after text rewriting using the GPT-3.5 turbo model. Specifically, we employed the *gpt-3.5-turbo-0613* version with the prompt *"Rewrite the following paragraph:"*.

Table 4 demonstrates that, despite not being specifically designed for rewrite robustness, our method showed significant resilience against text rewriting. After text rewriting using GPT-3.5, the detection F1 score of our method decreased by only an average of 6.7. Notably, our method outperformed the key-based detection approach in this robustness scenario, with an average improvement of 1.5 percentage points. Therefore, our watermarking algorithm not only facilitates our watermark detection but also exhibits strong robustness.

## C  TRAINING DETAIL OF WATERMARK GENERATION AND DETECTION NETWORK

In this section, we will provide a more detailed introduction to the training details of the watermark generation network and watermark detection network.

For the watermark generation network, the network details have been introduced in Section 4.2. Specifically, for all tokens within a window size, each token is first passed through an embedding network E to obtain its embedding. Then, all embeddings are concatenated and passed through a fully connected network, FFN, to produce classification logits:

$$\mathcal{W}(\boldsymbol{x}_{n-w+1:n}) = \text{FFN}(\text{E}(x_{n-w+1}) \oplus \text{E}(x_{n-w+2}) \oplus .... \oplus \text{E}(x_n)). \tag{10}$$

For the embedding network E, there are five layers, the input dimension is 16, and the output dimension of each intermediate layer, including the output layer, is 64. For the FFN, its input dimension is 64*w, where w is the window size. It is a three-layer fully connected network, with the first two layer having an output dimension of 64 and the last layer having an output dimension of 1. After passing through a sigmoid function $\sigma$, the binary cross entropy is used to calculate the loss, as shown below:

$$L_g = -\frac{1}{N} \sum_{i=1}^{N} \left[ y_i \cdot \log(\sigma(W(x_{n-w+1:n}))) + (1 - y_i) \cdot \log(1 - \sigma(W(x_{n-w+1:n}))) \right]. \tag{11}$$

In the subsequent training process, as shown in Section 5.6, we used 5000 randomly generated pieces of data with desired label rate and trained for 500 steps with a batch size of 32 using the Adam optimizer with a learning rate of 0.01.

Similarly, the structure of the watermark detection network has been introduced in Section 4.4. The input to this network is all the texts to be tested. Firstly, every token is processed through an embedding network $E$, identical in structure and shared in parameters with the embedding network in the watermark generation network. Following the embedding of each token, the sequence is passed through an LSTM network to produce the final classification results for detection. This LSTM comprises five layers, with an input dimension of 64 for each unit, an intermediate output dimension of 128, and a final output dimension of 1 for the output logits, obtained by passing the last unit's output through a linear layer and then through a sigmoid function, utilizing BCE loss to generate the detector's loss, expressed as follows:

$$L_d = -\frac{1}{N} \sum_{i=1}^{N} \left[ y_i \cdot \log(\sigma(D(x)_i)) + (1 - y_i) \cdot \log(1 - \sigma(D(x)_i)) \right]. \tag{12}$$

During training, 10,000 data points were used, generated by the watermark generation network as random ID sequences (token IDs), with an equal ratio of watermarked and non-watermarked data. The network was trained using the Adam optimizer, a learning rate of 0.01, for 80 epochs, and a batch size of 32, with batch size set to 64.

It's important to note that although an LSTM was chosen for the watermark detection network architecture, it does not imply that watermark detectors are limited to LSTM. This choice was made due to the simplicity and effectiveness of LSTMs, but other architectures capable of processing sequential inputs, like transformers, could also achieve excellent results.

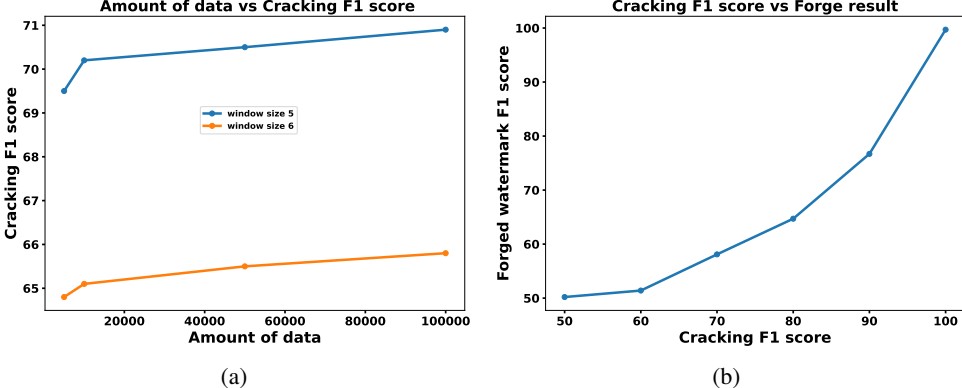

(a)                                  (b)

Figure 4: The left figure depicts the relationship between the different amount of data for training and the achievable cracking F1 score under a reverse training setting. The right figure demonstrates the effectiveness of watermark forgery at various cracking F1 scores.

## D   DETAILS OF REVERSE TRAINING

In Sections 4.5 and 5.4, we analyzed the challenges of training a watermark generation network using training data from a watermark detection network, both theoretically and experimentally. Here we elaborate on the intricacies of this reverse training process.

A key aspect is the definition of loss function for reverse training. Given a sentence's token list $\mathbf{x} = [x_1, ..., x_n]$, the detection network $\mathcal{D}$ assigns a label indicating watermarked (1) or not (0). We employ a new generative network, denoted as $\mathcal{W}$, to simulate the detection of watermarks. Specifically, this involves calculating whether each token contains a watermark and then computing the average:

$$\hat{p} = \frac{\sum_i^{T-w} \mathcal{W}(x_{i:i+w})}{T-w} \tag{13}$$

Utilizing the output of generative networks as labels, the final reverse training loss could be represented as follows:

$$L(\mathcal{W}) = -\mathcal{D}(\mathbf{x}) \log(\hat{p}) - (1 - \mathcal{D}(\mathbf{x})) \log(1 - \hat{p}) \tag{14}$$

For training, we used 10,000 random token lists of length 100-200, labeled by detector $\mathcal{D}$.

To further explore the impact of data amount to the reverse training attacks, we demonstrated in Figure 4(a) the changes in the cracking F1 score as the amount of data increases. It could be observed that, beyond a certain data amount, the cracking F1 score only experiences marginal improvements with the increase in data amount.

Moreover, to illustrate how different cracking F1 scores influence the degree of watermark forgery, we generated watermarked texts using watermarking rules with varying cracking F1 scores. These texts were then subjected to standard watermark detectors to calculate the corresponding detection F1 scores, as depicted in Figure 4 (b). We employed a watermark strength of $\sigma = 2$. The above result indicates that a cracking F1 score above 90 is required for successful watermark forgery.

## E   IMPACT OF WATERMARK ON MACHINE TRANSLATION TASK

To validate the impact of our watermarking method on text quality, we conducted further verification in the context of machine translation tasks.

Table 5: This table demonstrates the efficacy of our watermarking algorithm in machine translation tasks. We conducted experiments using four scenarios within the WMT14 dataset: English-French, French-English, German-English, and English-German. The machine translation model employed was NLLB-200-distilled-600M. We compared the watermark detection F1 score as well as the BLEU values before and after watermark insertion.

| Setting | Key-based F1 | Network-based F1 | Ori. BLEU | Wat. BLEU |
|---------|-------------|------------------|-----------|-----------|
| EN-FR   | 98.7        | 98.1             | 39.3      | 38.5      |
| FR-EN   | 99.2        | 98.9             | 37.9      | 34.0      |
| EN-DE   | 99.2        | 99.0             | 49.6      | 49.6      |
| DE-EN   | 99.4        | 99.1             | 38.5      | 37.9      |

Specifically, we utilized four scenarios from the WMT14 dataset Macháček & Bojar (2014): English-French, French-English, German-English, and English-German. The machine translation model employed was NLLB-200-distilled-600M Costa-jussà et al. (2022). We reported the model's BLEU scores without watermarking (Ori. BLEU) and with watermarking.

As indicated in Table 5, introducing watermarks only causes a minimal impact on BLEU scores, with an average decrease of only 1.3. In these scenarios, both key-based and our network-based watermarking detection methods achieved high detection F1 score. This further illustrates that our watermarking algorithm has only a negligible effect on text quality and model performance.

## F    EXAMPLE OF CYCLIC DOCUMENT

In Section 4.4, we introduced the approach of treating text as a cyclic document for watermark detection and training data construction. This method helps prevent a vulnerability related to the theft of watermark rules. We provide a more detailed explanation through a simple example in this section.

Consider a sentence $t$ with tokens $ABCDEF$. When the window size is 3, the labeling of each token is as follows: $C : \mathcal{W}(ABC) =$ watermarked, $D : \mathcal{W}(BCD) =$ not watermarked, $E : \mathcal{W}(CDE) =$ watermarked, $F : \mathcal{W}(DEF) =$ watermarked. However, without using a cyclic document, tokens A and B remain unlabeled, potentially creating a system vulnerability. For instance, if a user continuously alters the last token (here, F) and observes the detection confidence $\mathcal{D}(t = ABCDE?)$ of the network, a clear score distinction between watermarked and unwatermarked tokens emerges. If the user tests four characters and finds $\mathcal{D}(t = ABCDEA) \approx \mathcal{D}(t = ABCDEB) > \mathcal{D}(t = ABCDEC) \approx \mathcal{D}(t = ABCDED)$, it's easy to infer that with DE as a prefix, A and B are watermarked tokens, while C and D are not.

To address this vulnerability, we set the entire document as a cyclic document. We label the beginning token A as $\mathcal{W}(EFA)$ and B as $\mathcal{W}(FAB)$, ensuring each token has a corresponding label and thereby avoiding the aforementioned vulnerability.

## G    DETAILS OF DIRECT CRACKING ATTACK

Table 6: Occurrence frequencies of the most common (w-1)-grams under varying window sizes.

| Window Size | Most Common Prefix | Frequency |
|-------------|-------------------|-----------|
| 2           | *the*             | 0.04      |
| 3           | *. a*             | 0.018     |
| 4           | *âĠ Ļ s*          | 0.004     |
| 5           | *, âĠ Ļ Ġhe Ġsaid* | 0.00008  |

In this section, we detail another attack method mentioned in section 5.4: the direct cracking attack. This method, originating from the work of Sadasivan et al. (2023), does not require a watermark

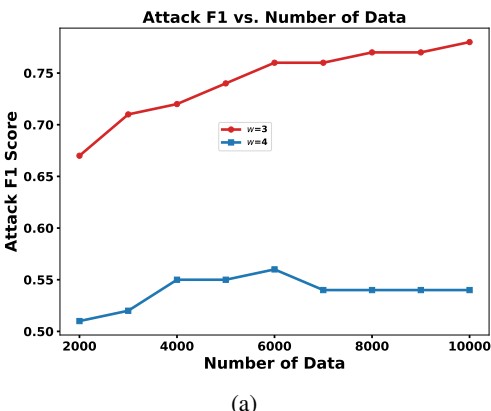

(a)

Figure 5: Variation in attack success rate with increasing data volume for window sizes of 3 and 4

detector. Instead, it solely analyzes changes in token frequency distributions. Specifically, with a window size of w, it examines the frequency the wth token appears given a prefix of length w-1 tokens. If a token's frequency significantly increases in the presence of a watermark, it is deemed to be part of the watermark token set.

As in Sadasivan et al. (2023)'s method, we examine the 181 most common (w-1)-token n-grams. Since even the most common (w-1)-grams become increasingly rare as w grows, predicting watermarked text also becomes increasingly difficult with this approach. As shown in Table 6, as the window size increases, the frequency of the most common prefixes decreases markedly. Analyzing this may require an extremely large amount of data, but through our analysis in Figure 5, when the window size grows to a certain extent, even increasing the data amount cannot significantly improve the effect.

## H   EXPANDING THE K IN TOP-K SAMPLING

Table 7: Resource consumption metrics of watermark generation network during 200-Token generation task under various k values and window sizes.

| K Value | Window Size | Computation Time (s) | Memory Consumption (MB) |
|---|---|---|---|
| 20 | 5 | 0.62 | 23 |
|  | 10 | 0.62 | 23 |
| 200 | 5 | 0.65 | 25 |
|  | 10 | 0.65 | 27 |
| 2000 | 5 | 0.67 | 46 |
|  | 10 | 0.70 | 48 |
| 20000 | 5 | 1.20 | 228 |
|  | 10 | 1.60 | 354 |

In Table 7, we comprehensively examined the impact of varying k-values on computation time and GPU memory utilization. All timings were recorded on a single V100 32G GPU. It can be observed that even when processing 20,000 tokens simultaneously, the computation time is only twice that of processing 20 tokens, while the memory consumption increases tenfold (with a window size of 5). This indicates that our algorithm demonstrates exceptionally low computational complexity with respect to increasing k-values.

## I   DETECTION F1 SCORE ACROSS DIFFERENT DOMAINS

To validate our claim in Section 4.4 that our watermark detection algorithm is robust to differences in text domain, we compare our network-based watermark detector to current key-based watermark detectors across various text domains in the DBPEDIA CLASS dataset (Table 8). Our algorithm

Table 8: Comparison of detection accuracy of our network-based watermark detector across text domains in the DBPEDIA CLASS dataset versus key-based watermark detectors.

| | Setting | FPR | FNR | F1 |
|---|---|---|---|---|
| **Domain: Agent** | Key-based Detector | 0.5 | 0.0 | 99.8 |
| | Network-based Detector | 0.0 | 0.7 | 99.6 |
| **Domain: Place** | Key-based Detector | 0.0 | 0.5 | 99.7 |
| | Network-based Detector | 0.1 | 2.4 | 98.9 |
| **Domain: Species** | Key-based Detector | 0.0 | 1.0 | 99.5 |
| | Network-based Detector | 0.0 | 2.8 | 98.3 |

achieves consistently high detection F1 score across domains, primarily because the random token ID lists used during training explore the space of $|V|^T$ possible inputs, covering all plausible texts. Thus, our detector can theoretically achieve similar detection F1 score across text domains.

