# OpenReview forum: "An Unforgeable Publicly Verifiable Watermark for Large Language Models"
_ICLR.cc/2024/Conference — ICLR 2024 poster_

### Official Review · Reviewer_kypk · 2023-10-24

**Soundness:** 3 good
**Presentation:** 2 fair
**Contribution:** 3 good
**Rating:** 6
**Confidence:** 2

**Summary:**

With the rapid development of Large Language Models (LLMs), these models are capable of generating human-like text, posing various risks such as the proliferation of false information on the Internet and the infringement of copyrights on creative works. Therefore, texts generated by LLMs need to be detectable and taggable. Existing watermarking algorithms are mostly public, requiring the detector to use the secret key employed in the watermark generation process, which exposes the method to vulnerabilities as attackers can easily remove and forge the text watermarks. To address these shortcomings, the authors propose a novel private watermarking algorithm designed specifically for LLMs. The approach uses two separate neural networks for watermark generation and detection, eliminating the dependency on a single secret key across both stages. The security of this method is enhanced through computational asymmetry, making it significantly more challenging to reverse-engineer the watermark generation algorithm from the detector. To reduce the complexity of training, the authors also introduced a technique to share token embedding parameters between the detector and the generator. Extensive experiments validate the efficacy of the proposed method, demonstrating nearly equivalent performance to public watermark algorithms while proving its resilience against two mentioned attacks.

**Strengths:**

1. The authors successfully identify the limitations inherent in existing public watermarking methods. They introduce a novel private watermarking approach that employs separate generation and detection models. This design transition from a public key to a private key significantly enhances the security of the watermarking process.
2. The authors offer a suitable explanation in Section 4.3, "Watermark Detection," to address the limitation of deep learning models in maintaining a static watermarked token ratio γ. The empirical validation of this explanation is provided in Section 5.6, "Watermark Generation Network Analysis."
3. While the performance of the proposed method under different datasets and models does not exceed that of public watermarking methods, the authors provide a plausible explanation for this discrepancy. Despite this, their method achieves competitive performance.
4. Regarding the strategy of shared embedding, the authors have validated its effectiveness through comprehensive ablation experiments in Section 5.3, "Analysis of Shared Embedding."

**Weaknesses:**

1. While the authors have identified the security flaws in public watermark methods, the discussion lacks specific references or examples of existing attack methods. To further bolster the superiority of their proposed approach, the authors might need to include comparative assessments against how public watermark methods fare against the two types of attacks discussed in Section 4.5, "Analysis of Privacy."
2. The last paragraph in Section 4.4, "Watermark Detection Network," regarding the construction of a cyclic document, is somewhat vague. A more detailed explanation, ideally supplemented with concrete examples, would enhance the clarity of this segment.

**Questions:**

1.& 2. The first two points are as mentioned in the weaknesses section.

3. It would be informative if the authors could elucidate the reasoning behind choosing LSTM networks for the Detection Network, especially when more advanced models are available.

---

> ### Author Response · Authors · 2023-11-14
>
> ## About the privacy of public watermark methods
>
> Thank you for your question.
>
> Firstly, please refer to first response in "Responses to all reviewers". We appreciate the concerns raised by reviewers seaW and CjAu regarding the definition of 'private watermark'. Indeed, we acknowledge the potential confusion in using this term to describe our method. Consequently, we have renamed our approach from 'A private watermark' to 'An Unforgeable Publicly Verifiable Watermark for Large Language Models'. It means that our proposed watermarking method could be publicly verified by a third-party detector, and even under this scenario, the watermark cannot be easily forged by an untrustworthy detector.  In other words, the previous concept "privacy" is interpreted more clearly as "unforgeablity" under third-party detection.
>
> Moreover, previous work that used shared key in both watermarking and detection process are referred to as key-based method, while our detection approach is named as network-based method. All related concepts in the updated version of our paper have been revised accordingly.
>
> Under this concept, the question "about the privacy of public watermark method" will be interpreted as "about the unforgeability of key-based methods". In a third-party detection scenario, the shared key used in key-based methods are exposed to the public.  Therefore, the key could be used to directly forge the watermark with no need of any attacks. That is, the key-based methods have no unforgeability.
>
> You might get confused about why we leverage spoofing attack and reverse engineering attack to test unforgeability of our watermarking method. That is because, by using our network-based detecting method, there is no watermark generation details (such as the watermark key) exposed to public. Therefore, if an attacker wants to acquire the watermark generation details and then use it to forge watermark, they could only try using sophisticated analysis such as spoofing attack and reverse engineering attack mentioned in our paper.
>
> We greatly appreciate your query and have clarified these issues in the updated version of our PDF.
>
> # About the cyclic document
>
> Consider a sentence $t$ with tokens $A B C D E F$. When the window size is 3, the labeling of each token is as follows: $C: \mathcal{W}(ABC) = \text{watermarked(green)}$, $D:  \mathcal{W}(BCD) = \text{not watermarked(red)}$, $E:  \mathcal{W}(CDE) = \text{watermarked(green)}$, $F:  \mathcal{W}(DEF) = \text{watermarked(green)}$. However, without using a cyclic document, tokens A and B remain unlabeled, potentially creating a system vulnerability. For instance, if a user continuously alters the last token (here, F) and observes the detection confidence $\mathcal{D}(t = ABCDE?)$ of the network, a clear score distinction between watermarked and unwatermarked tokens emerges. If the user tests four characters and finds $\mathcal{D}(t = ABCDEA) \approx \mathcal{D}(t = ABCDEB) > \mathcal{D}(t = ABCDEC) \approx \mathcal{D}(t = ABCDED)$, it's easy to infer that with DE as a prefix, A and B are watermarked tokens, while C and D are not.
>
> To address this vulnerability, we set the entire document as a cyclic document. We label the beginning token A as $\mathcal{W}(EFA)$ and B as $\mathcal{W}(FAB)$, ensuring each token has a corresponding label and thereby avoiding the aforementioned vulnerability.
>
> Thank you for your question. We have updated our PDF with the above content in its updated version (appendix F).
>
>
> # The selection of LSTM network
>
> The primary reason for selecting LSTM is that the input to the watermark detector is a variable-length sequence, and LSTM is a relatively classic RNN structure capable of processing such sequences. Although models like Transformer could be considered, we did not pursue further selection or optimization in this direction because the performance of LSTM was already satisfactory for our purposes. This lack of deliberate choice and optimization in network architecture also suggests that there is still room for improvement in our method.
>
>
> ## Reference
>
> Kirchenbauer, et al. "A watermark for large language models."

---

### Official Review · Reviewer_seaW · 2023-10-30

**Soundness:** 3 good
**Presentation:** 2 fair
**Contribution:** 3 good
**Rating:** 3
**Confidence:** 3

**Summary:**

This paper presents a new watermarking scheme for LLMs using two separate neural networks. The proposed watermarking method preserves the privacy of the secret key since watermark detection does not require the secret key used in watermark generation. The authors show via empirical results that the watermark detection method achieves a high detection rate.

**Strengths:**

The paper has the following strengths:
+ The authors propose a new private watermarking scheme that disentangles watermark generation and watermark detection. This addresses the privacy concern of requiring the secret key in watermark detection.
+ Empirical results show that it's difficult to reverse watermark generation from watermark detection, and also the proposed detection method achieves high detection rate.

**Weaknesses:**

The paper has the following weaknesses:
- The contribution of the private watermarking scheme is not clearly justified. If the designer wants to protect the secret key, he can use a public key encryption scheme to design the watermarking scheme. Particularly, they can use a secret key in watermark generation, and a public key for watermark detection. It's not clear why the designer has to use two neural networks for watermark generation/detection.
- There is no clear discussion about how watermark generation and the watermark detection model are trained. There is no explanation about the loss function used either. Alg.1 assumes watermark generation network W as input while it's not clear how W is trained in the first place.
- The results shown in Section 5 are not comprehensive. It's not clear whether watermark generation has negative impacts on the main task of the LLM and this is not measured. For existing DL watermarking schemes, there is a fidelity requirement on the watermark embedding which shall be measured as part of performance evaluation. Please consider define the criteria for LLM watermarking and report the results in the paper.

**Questions:**

Please address the weaknesses above.

---

> ### Author Response · Authors · 2023-11-14
>
> ## The contribution of the private watermarking
>
> Please refer to first response in "Responses to all reviewers".
>
> We acknowledge the potential confusion in using "private watermark" to describe our method. Consequently, we have renamed our approach from 'A private watermark' to 'An Unforgeable Publicly Verifiable Watermark for Large Language Models.' All related concepts in the updated version of our paper have been revised accordingly. Next, we will thoroughly explain the contribution of our "unforgeable publicly verifiable watermark".
>
> You have mentioned that a public key encryption algorithm could easily achieve the public verifiability. We assume the "public key encryption" refers to integrating methods like asymmetric encryption or digital signatures, which employ public-private keys into watermark algorithms. While these methods sound promising, its implementation in the context of large model watermarks is not straightforward. And most importantly, it is hard to design a public key encryption algorithm that can make the watermark unforgeable.
>
> For example, in a recent work, Fairoze et al. (published after the ICLR submission deadline) use digital signature technology to achieve public verifiability. However, in the watermark detection process, they still require extracting the watermark features first and then use the public key to verify them. Therefore, under a third-party public detection scenario, the extracted watermark features are exposed to public,  which can be directly used to forge watermark.
>
> Our method, however, has unique advantages. Users can only get from our detection network whether a text contains a watermark, without knowing the specific generation details of the watermark. Thus, while the method you propose is an admirable vision, integrating this idea into the watermarking domain is highly challenging, and currently, no feasible solutions exist.
>
> We believe our approach remains irreplaceable in the unforgeable publicly verifiable watermark domain, contributing uniquely to this field.
>
> We welcome further discussion on this topic.
>
>
> ## Training detail of watermark generation and detection network
>
> Thank you very much for your detailed question. Indeed, we did not elaborate on all the training details of the watermark generation and detection networks in the original text. However, we have thoroughly described the network architecture, input, and output in the original text. Moreover, the hyperparameters related to network training have already been introduced in  section 4.
> Overall, the training of both networks does not involve any particularly unique tricks. We will provide a detailed introduction here, and this information has already been added to the updated version of the PDF.
>
> Regarding the watermark generation network, firstly, for a token within a window size, an embedding network is used to generate its representation. This embedding network is a fully connected network. Then, all tokens within that window size are concatenated and passed through another fully connected network, which performs a binary classification. This involves first using a sigmoid function followed by binary cross-entropy loss. The formula can be represented as follows:
>
> $$ L_g = -\frac{1}{N} \sum_{i=1}^{N} \left[ y_i \cdot \log(\sigma(W(x_{n-w+1:n}))) + (1 - y_i) \cdot \log(1 - \sigma(W(x_{n-w+1:n}))) \right] $$
>
> For the watermark detection network, the process is quite similar. The representation of each token to be detected is generated using an embedding network. Then, all these representations are fed into an LSTM network. The output of this network undergoes a binary classification. The formula for this is:
>
> $$ L_d= -\frac{1}{N} \sum_{i=1}^{N} \left[ y_i \cdot \log(\sigma(D(x)_i)) + (1 - y_i) \cdot \log(1 - \sigma(D(x)_i)) \right] $$
>
>
> ## Impact of watermark on text quality
>
> Please refer to forth response in "Responses to all reviewers".
>
> ## Reference
>
> Kirchenbauer, et al. "A watermark for large language models."
>
> Fairoze, et al. "Publicly Detectable Watermarking for Language Models."

---

### Official Review · Reviewer_CjAu · 2023-10-31

**Soundness:** 3 good
**Presentation:** 3 good
**Contribution:** 2 fair
**Rating:** 6
**Confidence:** 4

**Summary:**

The authors propose a private watermarking method for large language models that allows the public verification of a watermark by using two verification keys, one of which can be released publicly. The authors propose an algorithm involving an LSTM to generate the watermarked text and show that their watermark is effective and difficult to reverse-engineer.

**Strengths:**

* The experimentation section is thorough, although it could be explained better

* The authors tackle an important and timely problem of watermarking language models.

* The idea of splitting the generation and verification watermarking key is interesting

**Weaknesses:**

**Unclear Definition of Private Watermarking**

Some statements are confusing. For example, on p1, the authors write the following.
> However, current watermarking algorithms are all public, which means the detection of watermarks requires the key from the watermark generation process.

But then, the authors state that any watermarking method can be made private by limiting who has access to this detection key (also p1).

> Although Kirchenbauer et al. (2023) have suggested that the watermark detection process could be placed
behind the web API to achieve the effect of private watermarking, this approach requires substantial
server resources and robust designs against hacking (even social engineering).

Hence, there is a trivial method to make watermarking algorithms private, meaning that not all watermarking methods are public. I am confused by the author's definition of a private and public watermark. I understand the general idea of separate keys for generating and verifying a watermark, but could the authors please elaborate on their definition of a private watermark? Why is their method not public (as the definition used by Kirchenbauer [A] suggests: A watermark is public if anyone can detect a watermark in text).

**Imprecise Language**

* What does "cracking the watermarking rules" mean? Are the authors referring to the integrity of their watermarks?
* What does it mean to "implement watermarking in a privacy-preserving manner"?
* What are "definite labels" (p9)?

**Robustness**

The authors' focus appears to be on the effectiveness and privacy of their watermark, and they do not study its robustness. An attacker who can verify a watermark in any text sequence should be much more capable of removing the watermark. Why did the authors not evaluate robustness? What good is a watermark that is not robust? Even if the watermark is not robust, I would appreciate it if the authors included these experiments showing their watermark's limitations.


**Minor Questions**

* What does FFN stand for in EQ 2?

* How many parameters does the LSTM have?

* In EQ 3, is "P" supposed to be the probability? Earlier, it was defined to be the logit score.

* In EQ 8, what is $f$?
--------
[A] Kirchenbauer, J., Geiping, J., Wen, Y., Katz, J., Miers, I., & Goldstein, T. (2023). A watermark for large language models. arXiv preprint arXiv:2301.10226.

**Questions:**

* Please clarify the motivation and definitions of "privacy". Why are existing watermarking methods not private?

* How robust is your watermark?

---

> ### Author Response · Authors · 2023-11-14
>
> ## Unclear Definition of Private Watermarking
>
> Please refer to first response in "Responses to all reviewers".
>
> The objective of the private watermark is to detect of the watermark without revealing watermark generation details.  This is valuable because if the key is exposed, others can forge the watermark.
>
> Although Kirchenbauer et al. (2023) claim that the key can be concealed within an API for subsequent services, this approach still relies on the generation key for detection, disqualifying it as a true private watermark algorithm.  Furthermore, it's important to note that hiding watermark generation details at the algorithmic level does not contradict to additional system design (API).
>
> A relatable analogy is with asymmetric encryption algorithms like RSA, where encryption is done with a private key, but decryption uses a public key. In contrast, symmetric encryption algorithms might employ a key for encryption and offer an API service using the same key for decryption, mimicking asymmetric decryption. However, this is not genuine asymmetric encoding, as decryption still requires the encryption key.
>
> Hosting the decryption or watermark detection process on servers introduces additional complications, such as costly API fees, the need to secure servers against hacking, and trust issues with the service provider. For instance, consider a scenario where an entity accused of generating inappropriate texts is brought to court. It would be unreasonable to rely on the same entity to verify the texts through watermark detection.
>
> We welcome further discussion on this topic.
>
> ## Explain the "cracking the watermarking rules"
>
> The watermarking rules essentially provides rules on how watermarking is generated.  For instance, in the algorithm of Kirchenbauer et al. (2023), watermarking is implemented by dividing the vocabulary into green (watermarked) and red (unwatermarked) lists, then increasing the probability of tokens from the green list at each generation step. The rule here refers to the partition of the green and red list. For example, in Kirchenbauer et al. (2023), for a window size of 2, determining which tokens follow the prefix token 'there' and belong to the green list constitutes a rule.
>
> 'Cracking the watermarking rules' fundamentally means inferring these specific rules directly, rather than obtaining the watermark's key.  For example, in Sadasivan et al. (2023), a spoofing attack could deduce a specific watermark rule by analyzing word frequency within watermarked texts. If a watermarking algorithm's rules can be easily deduced by such methods, it is essentially not a private algorithm.
>
> A good private watermark algorithm could not hide the generation key but also robust to the attacks that try to crack the watermarking rules.
>
> ## About the "implement watermarking in a privacy-preserving manner"
>
> Thank you for your question. "Privacy-preserving manner" refers to the ability to detect watermarks without needing the key used in the watermark generation process. Indeed, we acknowledge that the original statement was somewhat confusing. We have clarified this point in the revised PDF provided.
>
> ## What are "definite labels" (p9)?
>
> In our context, 'definite labels' refer to labels with high confidence levels, where the label here refers to whether a text contains a watermark. The corresponding z-scores for data with 'definite labels' significantly deviate from the threshold value used in the public z-score based detection.
>
> ## About robustness
>
> Please refer to third response in "Responses to all reviewers" and the table5 in the updated PDF version.
>
> ## What does FFN stand for in EQ 2?
>
> The term "FFN" here refers to a fully connected classification network. We apologize for any confusion caused and have provided a revised explanation in the updated version of the PDF.
>
> ## How many parameters does the LSTM have?
>
> Our LSTM only comprises five layers, with an input dimension of 64 for each unit, an intermediate output dimension of 128, and a final output dimension of 1 for the output logits.
>
> Therefore, the parameters of LSTM are negligible compared to those of large language models.
>
> ## In EQ 3, is "P" supposed to be the probability? Earlier, it was defined to be the logit score.
>
> P indeed represents probability, and P_n was previously used to denote a special probability symbol. However, considering potential confusion, we have now adopted L_n to represent logits. This change has been clarified in the updated version of our PDF.
>
> ## In EQ 8, what is f ?
>
> In this context, 'f' represents an ideal generation network, corresponding to the 'g' function mentioned in EQ9, where 'g' refers to an ideal detection network. Our aim here is to demonstrate that training 'g' using 'f' is significantly simpler than the reverse process of training 'f' using 'g', which forms the basis of our algorithm's privacy. Thank you for your question; we have addressed this point in the revised version of our PDF.

---

> > ### Comment · Reviewer_CjAu · 2023-11-15
> > **Thank you for your rebuttal.**
> >
> > Thank you for addressing my questions in your rebuttal. Your answers clarified the paper a lot! I still have some questions left.
> >
> > > The objective of the private watermark is to detect of the watermark without revealing watermark generation details. This is valuable because if the key is exposed, others can forge the watermark.
> >
> > I have some problems with this definition. First, it only specifies the objective of "private" watermarking, but does not define it. Second, it is potentially confusing, because the term "watermark generation details" is not clearly defined. What are watermark generation details? What are "watermark rules"? I do not want to sound pedantic, but this has confused me when I first read the paper and I think it will also confuse other readers.
> >
> > As stated in my review, Kirchenbauer et al.'s definitions make sense to me: A watermark is public if anyone can invoke the watermark detection functionality, and it is private if that access is restricted. The fact that an attacker cannot generate their own watermarked text is an interesting property of your method that arises from the use of two separate keys. In that sense, I still believe your watermark should be called a "public" watermark. Also, the paper that you linked by Fairoze et al. refer to their watermark as "public verifiable", which makes sense to me. They refer to a "public" and a "private" key, inspired by asymmetric encryption and their notation avoids confusion with the already existing term of "private watermarking".
> >
> > ## Threat Model
> >
> > I think the paper needs a proper threat model section describing the attacker's and defender's capabilities and goals. That would clarify the claims made. Does the attacker have access to similar performing open-source models? Is the attacker limited in the number of queries they can make to the watermarked model? When do you consider your method "broken", i.e., when does it lack robustness (attacker can remove the watermark) or integrity (attacker can generate watermarked text)? When do you consider two texts to be "different"? All these questions have to be answered in the paper.
> >
> > ## Robustness
> >
> > Thank you for providing results on the robustness of your method. It is interesting to see that your watermark has robustness, but in a public detectable watermark, an attacker has more capabilities than "just" paraphrasing it. A simple extension would be to extend this attack to "K" paraphrasings and then select the best paraphrase. I do not ask the authors to evaluate all these untested attacks, as this would likely extend beyond the scope of this work, but please add this to your limitation section.
> >
> > ## Distillation
> >
> > Regarding the size of the LSTM, I was wondering whether a small LSTM size makes the model susceptible to distillation-type attacks? Without using the public key, an attacker could sample the model N times and fine-tune a public model to generate watermarked data. Do you have any insights of this attack?

---

> > > ### Author Response · Authors · 2023-11-16
> > > **Thank you for your valuable feedback**
> > >
> > > ## Unclear Definition of Our Watermarking
> > >
> > > Please refer to new first response in "Responses to all reviewers".
> > >
> > > Thank you for raising your question. We have indeed recognize that the concept of a "private watermark" can be somewhat confusing. Consequently, we have renamed our methodology from "A private watermark" to "An Unforgeable Publicly Verifiable Watermark for Large Language Models." We have also revised all related concepts in the updated version of our paper.
> > >
> > > Current watermarking algorithms for large models share the same key during the generation and detection phases. This approach functions well when watermark detection is restricted to the watermark owner (private detection). However, if the detection process is made public, the watermark's key becomes exposed, leading to the risk of watermark forgery.
> > >
> > > Our watermarking algorithm, however, employs a neural network for detection, allowing public verification without revealing the watermark's generation method. This feature renders our watermark unforgeable and publicly verifiable.
> > >
> > > The implementation of an unforgeable publicly verifiable watermark presents significant challenges and, to date, there has been no other work achieved this goal. For example, Fairoze et al. (published after the ICLR submission deadline), use digital signature technology to avoid exposing the private key, they still require extracting the watermark features first and the use the public key to verify them. The extracted watermark features expose the watermark rules, which can be directly used to forge watermark.
> > >
> > > We greatly appreciate your valuable feedback and patient response. We welcome further discussion on this topic.
> > >
> > > ## About Threat Model
> > >
> > > Thank you for your valuable suggestion. We have provided a threat model section in Appendix G.
> > >
> > > ## About Robustness
> > >
> > > Thank you for your valuable suggestion. We have mention more robustness issue in the limitation part of conclusion section.
> > >
> > > ## About Distillation
> > >
> > > If we understand correctly, the distillation attack you proposed appears to be quite similar to the reverse training attack we have already employed in our paper. We have attempted to use a public LSTM detection network to distill a new watermark generator. This process is detailed in Figure 2(a).
> > >
> > > Thank you once again for your valuable feedback, which greatly contributes to the enhancement of our paper's quality.

---

> > > > ### Comment · Reviewer_CjAu · 2023-11-18
> > > > **Thank you for the changes.**
> > > >
> > > > Thank you for all the changes that you made to the paper. I believe it is much better already. I still have two concerns.
> > > >
> > > > ### Threat Model
> > > >
> > > > I read your threat model in Appendix - G, but it does not answer all of the questions that I raised. Please rewrite the threat model for clarity and ensure that all questions are given an answer. I think the threat model should go into the main part of the paper since it essentially defines the rules of the game. Could you include a proper threat model in Section 3 - Problem Definition?
> > > >
> > > > ### Distillation
> > > >
> > > > As far as I understand,reverse training is different from distillation. For distillation, I would expect to see a graph that shows the number of watermarked samples (obtained from the provider's model) versus the detection accuracy (for some z-score threshold). The attacker simply uses supervised fine-tuning to forge the watermark given many examples.
> > > >
> > > > ### Minor
> > > >
> > > > * Please, whenever you refer to the Appendix make sure to refer to the specific subsection.
> > > > * At this point, the number of changes applied to the paper gets to a point where I would encourage the authors to please highlight all changes in some color (not with latexdiff, but just highligh them). This will also make it easier for the other reviewers to see what has changed at a glance.

---

> > > > > ### Author Response · Authors · 2023-11-19
> > > > > **Thank you for your patient and valuable feedback**
> > > > >
> > > > > ## About Threat Model
> > > > >
> > > > > We regret that the description of the Threat Model in the previous version was still lacking. We have now moved the definition of the Threat Model to Section 3. At the same time, we provide a detailed response to your query here to reduce some misunderstandings.
> > > > >
> > > > > Firstly, regarding the goal of an attacker, there are two different attack scenarios. One involves attempting to remove watermarks by modifying the text, such as through rewriting or synonym replacement. In this scenario, we use the same setting as other current watermark algorithms, which assumes that the user cannot access the watermark detector. Otherwise, with sufficient effort, the watermark can always be eliminated. The second scenario occurs when the user can access the watermark detector; here, the user's query number is not limited, and the attacker tries to crack the watermark's generation method to forge it.
> > > > >
> > > > > For the first scenario, if a user can eliminate most of the watermark text using some rewriting algorithms, the watermark algorithm is considered 'broken.' In the second scenario, if a user cracks enough watermark rules to forge the watermark, then the watermark is also considered 'broken.'
> > > > >
> > > > > Regarding the question, 'When do you consider two texts to be "different"?' we are not entirely sure we fully understand your meaning. We can only define it in the context of watermarking: if two texts convey different meanings or are labeled differently under a watermark detector, they are considered different.
> > > > >
> > > > > Regarding the question, 'Does the attacker have access to similar performing open-source models?'. In the first scenario, attacker has no access to any models. In the second scenario, the attacker has access to the origin detector model, so no need for open-source models.
> > > > >
> > > > > ## Distillation
> > > > >
> > > > > I apologize, but we still haven't fully grasped the concept of distillation. The discussion here is based on the definition of the second attack scenario mentioned above, where a user has access to a watermarked detector. If the user already has access to the detector, why is there a need to distill another separate detector?
> > > > >
> > > > > According to your original description
> > > > > ```
> > > > > Without using the public key, an attacker could sample the model N times and fine-tune a public model to generate watermarked data.
> > > > > ```
> > > > > We understand that this public model can only be a watermark generator model. Therefore, we used the training data generated by the accessible detector to fine-tune a public generator, and we have termed this process 'reverse training'. It seems, according to your original intent, that the goal of Distillation is to produce a detector model. However, directly using a detector model does not effectively forge watermarks (our cyclic document mechanism effectively prevents this vulnerability, details are in appendix F).
> > > > >
> > > > >
> > > > > Additionally, regarding your question,
> > > > > ```
> > > > > I would expect to see a graph that shows the number of watermarked samples (obtained from the provider's model) versus the detection accuracy (for some z-score threshold).
> > > > > ```
> > > > > We have further supplemented the effects of reverse training using different data amount. Here, we compared the results of reverse training using different data amount under a window size of 5 and 6 in figure4(a) and appendix D.
> > > > >
> > > > > It can be observed that even with a substantial increase in the amount of data, the effectiveness is only slightly better than the cracking results we originally achieved with 10,000 data. Moreover, regarding the cracking F1 score, generally, an accuracy rate of over 90% is required for effective watermark forgery. Our experimental results are as follows: under the same watermark strength of $\sigma=2$, the detection accuracy rates of watermarked texts generated with different Cracking F1 scores are shown in figure4(b) and appendix D.
> > > > >
> > > > > ## Minor
> > > > >
> > > > > Thank you for your suggestion. We have highlighted the revised sections in our paper and made adjustments to the specific references in the appendix.
> > > > >
> > > > > Finally, we would like to extend our gratitude for your patient advice, which has been very beneficial in enhancing the quality of our work.

---

> > > > > > ### Comment · Reviewer_CjAu · 2023-11-22
> > > > > > **Thank you for your reply**
> > > > > >
> > > > > > Thank you for all these modifications. I like the paper and idea and will increase my score.

---

### Official Review · Reviewer_PMgN · 2023-11-01

**Soundness:** 2 fair
**Presentation:** 3 good
**Contribution:** 2 fair
**Rating:** 6
**Confidence:** 3

**Summary:**

This paper presents a method for LLM watermarking. It can handle the security breaches and counterfeiting problems of existing LLM watermarking methods caused by secret key, through using two different neural networks for watermark generation and detection. Experiments on two datasets verify the effectiveness of the proposed method.

**Strengths:**

1. LLM watermarking is a very important problem.

2. The proposed method can handle the security breaches and counterfeiting problems of existing LLM watermarking methods.

3. The proposed method is effective according to the reported results.

**Weaknesses:**

1. Please correct me if I am wrong. It seems that there is no comparison with baseline methods in terms of both watermarking effectiveness and text generation quality.

2. It is not clear whether text edit methods like paraphrase can make the proposed method invalid.

**Questions:**

Are your methods robust to text edit methods like paraphrase?

---

> ### Author Response · Authors · 2023-11-14
>
> ## About Baseline Comparison
>
> Related responses: please refer to the second and forth responses in "Responses to all reviewers".
>
> The key-based detection method in Table1 is the same to the KGW watermark algorithm by Kirchenbauer et al. (2023).
>
> This implies that Table 1 compares the watermark detection accuracy of Kirchenbauer et al. (2023) (denoted as 'key-based') with our unforgeable watermark algorithm.  And due to the lack of alternative unforgeable watermark algorithms, we did not conduct direct comparisons with other unforgeable watermark methods.
>
> Moreover, since our watermark generation method is almost identical to that of Kirchenbauer et al. (2023), the impact on text quality should be exact the same to Kirchenbauer et al. (2023).
>
> Despite this, we have demonstrated in Figure 2(b) the variation of detection F1 score and PPL with changes in the $\delta$ parameter. In our method, the $\delta$ hyperparameter is set to 2, indicating that our approach achieves high detection accuracy without significantly affecting text quality.
>
> We have also conducted analysis of text quality changes in machine translation task, affirming that our watermarking method does not significantly impact text quality. More details in the forth response in "Responses to all reviewers".
>
> The updated PDF version now includes additional clarifications and descriptions.
>
> ## About robustness to paraphrase attack
>
> Please refer to third reply in "to all reviewers" and the table4 in the updated PDF version.
>
> ## Reference
>
> Kirchenbauer, et al. "A watermark for large language models."

---

### Author Response · Authors · 2023-11-14
**Responses to all reviewers**

# Responses to all reviewers.

## 1. About the definition of our watermark.

We appreciate the concerns raised by reviewers seaW and CjAu regarding the definition of 'private watermark'. Indeed, we acknowledge the potential confusion in using this term to describe our method. Consequently, we have renamed our approach from 'A private watermark' to 'An Unforgeable Publicly Verifiable Watermark for Large Language Models.' All related concepts in the updated version of our paper have been revised accordingly.

Current watermarking algorithms for large models share the same key during the generation and detection phases. This approach functions well when watermark detection is restricted to the watermark owner (private detection). However, if the detection process is made public, the watermark's key becomes exposed, leading to the risk of watermark forgery.

Our watermarking algorithm, however, employs a neural network for detection, allowing public verification without revealing the watermark's generation method. This feature renders our watermark unforgeable and publicly verifiable.

The implementation of an unforgeable publicly verifiable watermark presents significant challenges and, to date, there is no other work achieved this goal. For example, Fairoze et al. (published after the ICLR submission deadline), use digital signature technology to avoid exposing the private key, they still require extracting the watermark features first and the use the public key to verify them. The extracted watermark features expose the watermark rules, which can be directly used to forge watermark.


## 2. About the baselines

It should be noted that in the Table 1, the term "Key-based detection algorithm" is the same to the detection method of KGW  Kirchenbauer et al. (2023). Therefore, our work includes a baseline comparison with other methods. And due to the lack of other unforgeable watermark algorithms, we do not conduct comparisons with other unforgeable watermark methods.

## 3. About the robustness

The table below displays the detection results after rewriting the generated text using GPT-3.5.

| Model    | Window Size | Setting | F1(No Attack)| F1(Rewrite w. GPT 3.5) |
| -------- | ----------- | ------- | --------- | ------------------ |
| GPT2     | 1           | Key-based  | 100.0     | 96.4               |
| GPT2     | 1           | Network-based | 99.7      | 96.4               |
| GPT2     | 2           | Key-based  | 100.0     | 92.8               |
| GPT2     | 2           | Network-based | 99.5      | 96.2               |
| OPT 1.3B | 1           | Key-based  | 100.0     | 92.0               |
| OPT 1.3B | 1           | Network-based | 100.0     | 90.3               |
| OPT 1.3B | 2           | Key-based  | 100.0     | 90.1               |
| OPT 1.3B | 2           | Network-based | 99.3      | 91.8               |
| LLaMA 7B | 1           | Key-based  | 100.0     | 89.0               |
| LLaMA 7B | 1           | Network-based | 100.0     | 89.2               |
| LLaMA 7B | 2           | Key-based  | 100.0     | 87.2               |
| LLaMA 7B | 2           | Network-based | 98.6      | 92.9               |

As observed from the above table, after rewriting the text using GPT-3.5, the F1 score for detection of our method only decreased by an average of 6.7. Furthermore, our method outperforms key-based methods in this robustness scenario, averaging 1.5 percentage points higher. Thus, our watermarking algorithm not only achieves unforgeable watermark detection but also demonstrates excellent robustness. More details are shown in the Table4 in updated PDF.

## 4. About the Impact on Text Quality

We have shown in Figure 2(b) that our watermarking has a minimal impact on PPL.

To further demonstrate the negligible impact of our watermark on the text quality, we also show results of machine translation task here.

Specifically, we utilize the WMT14 dataset, with the machine translation model being NLLB-200-distilled-600M [Costa-jussà et al. (2022)]. We report the BLEU scores of the model without watermarking and with watermarking.

| Setting | Key-based F1 | Network-based F1 | Ori. BLEU | Wat. BLEU |
| ------- | --------- | ---------- | --------- | --------- |
| EN-FR   | 98.7      | 98.1       | 39.3      | 38.5      |
| FR-EN   | 99.2      | 98.9       | 37.9      | 34.0      |
| EN-DE   | 99.2      | 99.0       | 49.6      | 49.6      |
| DE-EN   | 99.4      | 99.1       | 38.5      | 37.9      |

The table show that our method only has a slight effect on BLEU scores, with an average decrease of only 1.3 in BLEU, further illustrating the minimal impact of our watermarking algorithm on text quality and model performance.

## Reference

Kirchenbauer, et al. "A watermark for large language models."

Fairoze, et al. "Publicly Detectable Watermarking for Language Models."

Costa-jussà, et al. "No language left behind: Scaling human-centered machine translation."

---

### Meta-Review · Area_Chair_vQPR · 2023-12-07

**Metareview:**

The paper presents a novel approach to private watermarking for text generated by large language models (LLMs). Reviewers' feedback highlights a few key areas of concern, including the lack of comparison with baseline methods, clarity in defining private watermarking, and the robustness of the proposed watermarking technique. There's also a call for more detailed discussions on the training of watermark generation and detection models, as well as the impact of watermark generation on the main task of the LLM. The authors have responded to these concerns by adjusting the definition of private watermarking, providing additional explanations, and supplementing their experiments. While there seems to be some disagreement among reviewers about the paper's quality, the authors have addressed the key issues raised.

**Justification For Why Not Higher Score:**

Firstly, there is a lack of comparative analysis with existing watermarking methods, leaving the effectiveness and efficiency of the proposed method unverified against established benchmarks. Secondly, the paper initially suffers from unclear definitions and imprecise language, particularly in defining private versus public watermarking. Though the authors have revised their definitions, this initial vagueness indicates a need for clearer conceptual grounding. Lastly, the robustness of the watermark against potential attacks and its impact on the primary functionality of LLMs are not thoroughly explored, which are critical aspects to consider for a watermarking method.

**Justification For Why Not Lower Score:**

The paper  introduces an innovative approach to watermarking in the context of LLMs, addressing privacy concerns inherent in public watermarking methods. The authors have made significant efforts to clarify their definitions and methodology in response to the reviewers' feedback, showing responsiveness and a commitment to improving their work. Additionally, the inclusion of additional experiments and detailed explanations in the appendix demonstrates the authors' attempt to comprehensively address the concerns raised. This responsiveness, coupled with the paper's contribution to a relatively underexplored area of LLM watermarking, justifies maintaining its current score.

---

### Decision · Program_Chairs · 2024-01-16

Accept (poster)